# Confocal Laser Endomicroscopy for Bladder Cancer Detection: Where Do We Stand?

**Angelo Naselli, Andrea Guarneri and Giacomo Maria Pirola** *

Department of Urology, San Giuseppe Hospital, Multimedica Group, 20123 Milano, Italy
* Correspondence: gmo.pirola@gmail.com

**Featured Application: Insight on actual evidence of confocal laser endomicroscopy applications for bladder cancer detection.**

**Abstract: Introduction:** Confocal laser endomicroscopy (CLE) is a relatively new technology that allows for a real-time in situ microscopic characterization of tissue lesions, being able to discriminate between low- and high-grade ones. After a first period of slow diffusion caused by technological limitations and elevated costs, CLE applications are rapidly spreading in different branches of medicine, and there is mounting evidence of its advantages for the management of different tumors such as bladder cancer (BCa), from both a diagnostic and a clinical point of view. In this systematic review (SR), we evaluate the state-of-the-art CLE for BCa management. **Material and methods:** We performed an SR and quality assessment analysis of the current literature in this regard following the PRISMA guidelines. All data were independently verified by two different authors and discrepancies were solved by a third author. Moreover, a quality-assessment analysis according to QUADAS-2 criteria was performed to evaluate the studies selected for SR. **Results:** A total of 158 articles were retrieved; of which 79 were rejected and 38 were removed as duplicates. After article selection, seven prospective studies were assessed for data extraction. These accounted for 214 patients overall, with a correspondence rate between CLE and histopathological examination ranging from 54.6 to 93.6%. Regarding quality assessment, three out five prospective studies have at least a high risk of bias in one QUADAS-2 domain, whereas the applicability always has a low risk of bias. **Conclusion:** Despite actual technical limitations, the preliminary results of this appealing technology are encouraging and should prompt further investigations.

**Keywords:** non-muscle invasive bladder cancer; cystoscopy; confocal laser endomicroscopy; endourology

## 1. Introduction

With the increasing augmentation of life expectancy and smoking habits, bladder cancer (BCa) represents the fourth most common neoplasm in men [1], occurring in 9 out of 100,000 men and in 2.2 out of as many women worldwide, representing the second most common malignant disease of the urinary system after prostate cancer [2]. As is well known, the first symptom is mostly the presence of gross hematuria, even if sometimes only urinary storage symptoms are present (i.e., pollakiuria) mimicking a urinary tract infection. The macroscopic aspects of the lesions at cystoscopy are variable, from sessile to flat ones, and sometimes BCa can manifest as hyperemic spots on bladder mucosa, mimicking cystitis; therefore, accurate diagnostic imaging is of outmost importance to discriminate the presence of BCa.

At the first presentation, BCa is non-muscle-invasive in 70% of cases (NMIBC), with possible stages Ta, T1, or carcinoma in situ (CIS), and muscle-invasive in the remaining 30% of cases. The surgical approach consists in transurethral endoscopic resection of the

bladder tumor (TURBT) for NMIBC and in radical cystectomy with eventual neoadjuvant chemotherapy for invasive forms [3].

Thus, improving TURBT quality and completeness is of outmost importance to reduce the risk of disease recurrence and progression as this clinical step is crucial for the histologic determination of cancer lesions and can prevent the risk of disease recurrence or progression. Even with enhanced fiber optics and digital visualization technologies, several reports outline the persistence of remnant neoplastic tissue after TURBT, which arrives in the 33–76% of cases, usually located at the border of the resected tumor or hidden as a flat lesion [4,5]. For this reason, current guidelines recommend a second procedure (re-TURBT) 4–6 weeks after the primary in cases of macroscopic incomplete cancer removal, pT1 stage or high-grade cancers, or in cases when muscular tissue is not present in the specimen [5].

To reduce the likelihood of incomplete resection, new diagnostic tools have been purposed and validated, demonstrating a clear benefit in improving the outcome of the first TURBT. Those can be divided into "macroscopic" techniques, aiming to enhance the different features of the neoplastic tissue in comparison to the normal one (i.e., increased vascularization or increased cell metabolism), and "microscopic" techniques, which directly allow the clinician to evaluate in real time the cellular architecture of the examined area.

Despite the current use and validation in clinical practice of contrast enhancement techniques like narrow-band imaging (NBI) and photodynamic detection (PDD) of bladder tumors using blue-light cystoscopy, there is still a quote of false-positive imaging related to those techniques, as they do not certainly discriminate between inflammatory and neoplastic "flat" areas [6]. To overcome this limit, an in vivo reading of microscopic cell architecture to directly discriminate between normal and neoplastic tissue has been advocated. Among them, confocal laser endomicroscopy (CLE) is the most promising tool that has already shown clinical benefit in clinical practice for gastrointestinal cancer detection (esophageal, stomach, and colon cancer), facilitating the discrimination of tissue dysplasia and carcinoma in situ [7,8]. Using a laser light source and fluorescent contrast dye, CLE has the potential property to allow a detailed and real-time understanding of tissue cell composition, allowing clinicians to postulate between areas with normal tissue and low-grade or high-grade tumors [9].

Aim of this systematic review (SR) is to focus on the existing literature concerning CLE applications in BCa management, reporting the actual validation studies of this methodic, its limitations, and future potentials.

## 2. Materials and Methods

### 2.1. Research Strategy and Selection Criteria

We performed systematic PubMed research in August 2022 according to the preferring reporting items for systematic reviews and metanalysis (PRISMA) guidelines [10] (Figure 1). The following criteria were used for article screening: (bladder cancer) and (confocal microscopy); (bladder cancer) and (confocal laser endomicroscopy); (bladder cancer) and (CLE); (bladder cancer) and (confocal endomicroscopy); and (bladder cancer) and (fibered confocal microscopy). Initially, titles were screened to identify eligible articles followed by the screening of abstracts. Finally, full text articles were analyzed, and relevant references were detected and included in the research. Moreover, individual searches were performed on PubMed, Google Scholar, Scopus, ScienceDirect, and SpringerLink. Reference lists of the selected articles were also searched, and additional studies were included. All authors approved the formulation of the search strategy and the article selection. Editorials, reviews, opinions, debates, case reports, and letters to editors were excluded from SR. We investigated the advantages, limitations, and usefulness of CLE in the diagnosis and management of BCa. GMP and AG performed the search and selection independently. All data were independently verified by two authors. In case of discrepancies, it was resolved by a discussion among the three authors (GMP, AG, and AN).

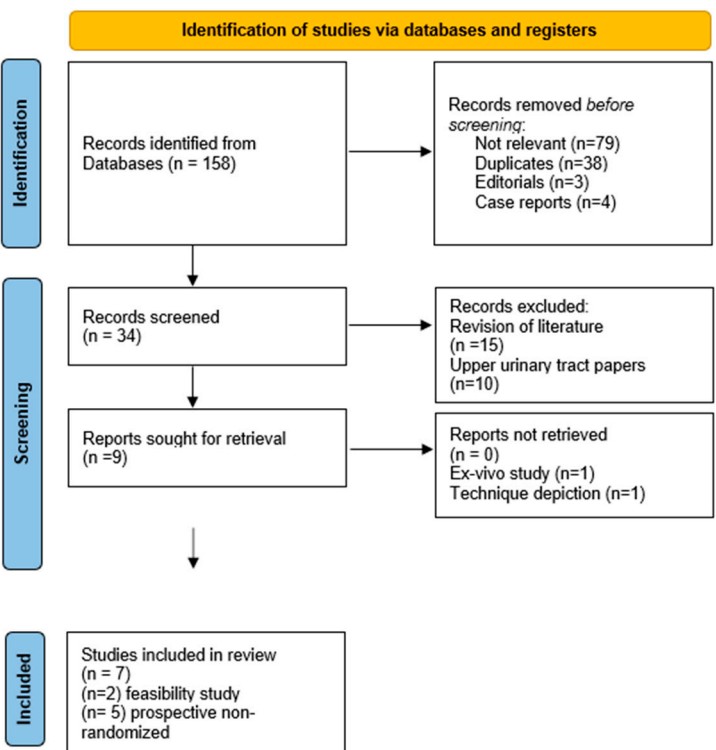

**Figure 1.** PRISMA study flowchart.

### 2.2. Quality Assessment and Data Extraction

A quality assessment of the selected non-feasibility studies was carried out through the quality assessment of diagnostic accuracy studies-2 (QUADAS-2) tool [11]. The patient setting is "cases undergoing TURBT". Index test is pCLE. The reference standard is the histopathological examination. The target condition is bladder cancer. GMP and AG independently scored the studies by strict and unconditional adhesion to the list of signaling questions [11]. In case of discrepancies, it was resolved by discussion among GMP, AG, and AN.

A full analysis of retrieved articles has been practiced to evidence all clinical data regarding patients' characteristics and diagnostic accuracy of CLE. We focused on the type of CLE and probe; contrast administration method; number of enrolled patients; mean patients' age (±SD); number of lesions; sensibility and specificity; and negative and positive predictive values of CLE in comparison with histologic diagnosis.

### 2.3. Confocal Microscopy Instrumentation, Technique, and Image Interpretation

The in vivo tool for CLE of bladder lesions is the Cellvizio clinical system (Cellvizio 100 series; Mauna Kea Technologies, Paris, France). The device is formed by a computer with dedicated software for image processing, a fiber-optic imaging probe for image acquisition (with variable diameter from 1.4 mm to 2.6 mm), and a laser scanning unit. The probe is delivered into the bladder through the working channel of the rigid or flexible cystoscope. Fluorescein, a contrast agent already adopted for other applications in human tissues [12], is used to stain the extracellular matrix, allowing the visualization of suspicious lesions. The contrast can be delivered intravenously or through topical bladder instillation. The confocal image is acquired in real time during the contact of the probe with urothelium and delivered to the screen through video sequences at 12 frames per second and stored in the system.

Wu et al. first depicted CLE imaging in a dedicated paper [13] and then implemented it in their subsequent study. The appearance of urothelial mucosa suddenly appeared easy to compare to the histological one. Normal bladder urothelium showed a uniform layer

of polygonal umbrella cell layer, then a smaller intermediate cell layer with a capillary network with erythrocytes in the lamina propria. Due to the small penetration depth of the confocal probe, the muscular layer cannot be appreciated from the surface and can only be visible in case of resection of epithelium (i.e., by contact of the probe with the tumor resection bed).

Low-grade superficial tumors appear as densely packed urothelial cells that are homogenous and monomorphic. Other characteristics include an increased amount of cellular papillary structures and fibrovascular stalks from tumor neoangiogenesis, not observed in normal or inflamed areas.

High-grade tumors show densely packed cells with pleomorphic population and irregular shape and loss of cellular cohesiveness, indistinct cell borders, and disorganized vasculature.

The most challenging lesion to recognize is represented by CIS. In fact, its appearance of larger cells with indistinct cell borders and extensive acellular areas are like inflammatory conditions. However, CIS cells appear larger and more pleomorphic even if this is not always clear. Moreover, inflammation presents loosely arranged aggregations of monomorphic cells in the lamina propria consistent with local recruitment of immunitary cells.

## 3. Results

A total of 158 articles were retrieved, of which 79 were rejected and 38 were removed as duplicates. From the remaining 34 papers, 15 were revisions of the literature and 10 articles were related to urothelial carcinoma of the upper urinary tract. Therefore, nine papers were selected. Among them, one was excluded [14] as it was an ex vivo study on radical cystectomy specimens; another one [13] was not considered as it presented a depictional atlas of CLE imaging obtained from 66 patients. Seven original articles were included in this SR [15–21]. Those include five prospective clinical trials with 214 patients overall and 312 bladder lesions where CLE was tested. The detailed PRISMA flowchart is reported in Figure 1. Regarding quality assessment, three out five prospective studies have at least high risk of bias in one Quadas-2 domain, whereas applicability has always a low risk of bias (Table 1). Moreover, the study design is different and there are no uniform comparators among all studies. The Cellvizio probe (Cystoflex UHD-R probe, Mauna Kea Technologies, Paris, France) was adopted in all studies, with fluorescein as contrast dye. Fluoresceine was administered intravenously or by bladder instillation. Sonn et al. [15] reported the advantage of the one way over the other. Overall, the correspondence rate between CLE and histopathological examination ranged from 54.6% to 93.6%. The work by Chang et al. [16] outlined that the sensitivity and specificity of CLE was strictly related to observer experience. In fact, the percentage of agreement between observers for BCa diagnosis ranged from 90% for experienced CLE urologists to 77% for nonclinical researchers. The same for CLE specificity, which ranged from 88% to 73%. Significative differences were also reported for cancer grading, with specificity ranging from 94% to 73% for low-grade lesions and from 64% to 60% for high-grade ones. However, all the retrieved articles outlined a high inter-observer variability, which cannot allow a clear depiction of CLE reliability for BCa diagnosis in this actual form. The results of the retrieved studies are schematically reported in Table 2.

**Table 1.** Risk-of-bias assessment according to Quadas-2 criteria [11].

| Publication | Patients' Selection | | Index Test | | Reference Standard | | Flow and Timing |
|---|---|---|---|---|---|---|---|
| | Risk of bias | Applicability | Risk of bias | Applicability | Risk of bias | Applicability | Risk of bias |
| Lucas 2019 [17] | Low | Low | High | Low | High | Low | Low |
| Wu 2019 [18] | Low | Low | Low | Low | Low | Low | Low |
| Lee 2019 [19] | Low | Low | High | Low | Low | Low | Low |
| Liem 2020 [20] | Low | Low | Low | Low | Low | Low | Low |
| Beji 2021 [21] | High | Low | Unclear | Low | High | Low | Unclear |

**Table 2.** Characteristics of included studies. CLE: confocal laser endomicroscopy; EV: intravenous contrast administration; BI: bladder contrast instillation; PPV: positive predictive value; NPV: negative predictive value; SD: standard deviation; NR: not reported.

| Author, Year of Publication [Ref] | Type of Study | Cellvizio Probe and Penetration Depth | Contrast Delivery Method (n) | Enrolled Patients (n) | Median Age (Range) | Overall Lesions (n) | Inter-Observer Agreement CLE Images | Histology-CLE Correspondence |
|---|---|---|---|---|---|---|---|---|
| Sonn, 2009 [15] | Feasibility | 2.6 mm–60 µm | EV (10) BI (5) Both (12) | 27 | 73 (range 47–90) | NR | NR | NR |
| Chang, 2013 [16] | Feasibility | NR | NR | NR | NR | 31 | -Experienced CLE urologists 90% -Novice CLE urologists 77% -Pathologists 81% | NR |
| Lucas, 2019 [17] | Prospective | 2.6 | BI | 53 | NR | 72 | Software-based interpretation | PPV: 74% NPV: 88% |
| Wu, 2019 [18] | Prospective | 2.6 mm | BI | 21 | 61 (32–81) | 21 | NR | 81% |
| Lee, 2019 [19] | Prospective | 2.5 mm | BI | 75 | 68.32 (±9.45 SD) | 119 | NR | PPV: 93.6% NPV: 68% |
| Liem, 2020 [20] | Prospective | 2.6 mm–65 µm | BI | 53 | 70 (62–79) | 66 | 76% | 70% |
| Beji, 2020 [21] | Prospective- pilot study | 2.6 mm | EV | 12 | 74 (52–94) | 34 | 73,5% | PPV: 54.6% NPV: 82.3% |

## 4. Discussion

Since the last two decades, the impact of new endoscopic technologies has markedly changed the diagnostic and management of BCa. For example, the recent advent of digital optics has outclassed the fiber-optic system for the enhanced quality of image definition, with relevant impact on BCa detection rate.

As outlined by several articles [4,5], the residual tumor rate after the first TURBT is not negligeable. To amend that, according to current guidelines [22], it is always advised to perform a second intervention (re-TURBT), not only when the first resection results are incomplete or in the absence of muscle layer in the specimen but also in all pT1 and high-grade cases. This procedure, even if mandatory, adds morbidity to the patient and potentially increases surgical waiting lists. Moreover, several articles show the actual limits of "white-light" (WL) cystoscopy, which can miss a consistent number of flat lesions—estimated around 58–68% [23,24]. To overcome those limitations, recent technological advances have been purposed and validated. The most adopted are actually the "macroscopic" ones, which enhance tissue vision through better visualization of tissue vascularization or by gaining better contrast between different aspects of the urothelium (i.e., the Storz professional image enhancement system (SPIES) or NBI) or to allow the detection of an area with incremented cellular metabolism (i.e., PDD, also known as "blue-light" cystoscopy) after the injection or instillation of photosensitizing agents (mainly hexylaminolevulinate) that bind to hyper-metabolic areas; those agents are sensible to a particular light spectrum (i.e., blue light) that emphasizes those areas during cystoscopy. However, all those techniques still have limitations as they cannot clearly discriminate between neoplastic and inflammatory areas. Therefore, rising attention is now given to "microscopic" techniques, which allow a real-time detection of tissue components and cellular aspects [25].

Although histology is the cornerstone for cancer diagnosis, this information is not available in real time during surgical intervention. Therefore, new imaging technologies that allow a direct characterization of cell architecture appear to overcome the limits of WLC and this is strongly demanded to reduce the risk of incomplete tumor resection or understaging [21,22].

The first purposed one is optical coherence tomography (OCT), which provides real-time cross-sectional images of tissues by means of infrared-spectrum microwaves, producing images of 2 mm in depth and 10–20 µm spatial resolution [25]. With OCT it is possible to discriminate between healthy and pathological urothelium and to postulate on the invasiveness of a bladder lesion. However, this technique provides no information about tumor grade [26].

Differently from OCT, CLE allows high-resolution optical imaging of microscopic tissue architecture with a 500–1000× magnification, a sort of "optical biopsy" that can give real-time information, allowing the clinician to discriminate the tissue microstructures of single cells, aiming for the distinction between low-grade and high-grade BCa areas and to ensure the complete asportation of nonvisible tumors. This can have an important impact to ensure the completeness of TURBT and to facilitate patients' follow-ups, as it can dramatically reduce the false-negative cystoscopy rate. Moreover, the actual availability of a validated image atlas [14] allows clinicians to provide a uniform interpretation of CLE imaging, therefore standardizing the technique and allowing the creation of dedicated formation programs and specific on-line courses.

The advantages of CLE have already been validated and confirmed in clinical practice in other specialities, such as in gastroenterology for colorectal polyposis/cancer discrimination, in superficial gastric dysplasia/cancer, and in Barrett's esophagus [27,28]. Moreover, CLE use has already been purposed and validated for upper-tract urothelial carcinoma (UTUC) diagnosis, showing an elevate correspondence with histopathological results [29]. This is of outmost importance as radical nephroureterectomy is the standard of care for UTUC, despite its clinical morbidity. Moreover, up to 25% of ureteric biopsies are not diagnostic according to the literature, making the follow-up after conservative treatment

difficult to carry on [30]. Therefore, a reliable tool that can discriminate between low- and high-grade lesions can have an important impact in clinical management, with a clearer identification of patients who can benefit from a conservative treatment.

According to the presented data, CLE application on bladder tissue appears feasible and effective; however, some aspects still need to be implemented.

### 4.1. CLE Feasibility

The first reported studies investigated the safety and technical feasibility of CLE for BCa detection. The technique appeared completely safe, without any reported adverse events among all studies.

Since the first report by Sonn et al. [15], CLE showed a definite image reporting of normal and neoplastic tissue, with macroscopic distinction between low-grade and high-grade cellular differentiation. In this study, the authors compared the results obtained by intravenous fluoresceine administration and bladder instillation, showing the potential advantages of fluoresceine instillation as being capable to detect bladder tissue and vascularization similar to the intravenous technique but with longer duration and with the possibility to modulate contrast intensity with the control of bladder irrigation. The subsequent five prospective studies reported a substantial similar percentage of inter-observer agreement for CLE image interpretation, which becomes more similar between experienced readers. Chang et al. [16] performed an inter-observer comparison study between experienced CLE urologists, non-experienced CLE urologists, a pathologist, and researchers. After two hours of computer-based training, in which participants were instructed to identify microarchitectural (flat vs. papillary, and tissue organization and vascularity) and cellular features (morphology, cohesiveness, and cellular borders) of benign and pathologic urothelium, they had to classify a series of 31 CLE video sequences consisting of 12 benign, 9 LG, and 10 HG images. The percentage of agreement was 81.6% for both microarchitectural and cellular features, showing that non-experienced readers were also able to interpret CLE images with proper teaching. The diagnostic accuracy and agreement improved with experience (i.e., 75% sensitivity and 64% specificity for experienced readers versus 46% sensitivity and 74% specificity for non-experienced ones). Liem et al. [20] then validated and improved the imaging interpretation criteria in 2018, giving as the main differential features for the definition of lesions and tumor grade the organization of cells, cellular morphology, and definition of cell borders. Lucas et al. [17] demonstrated the elevated accuracy of computer-based classification of CLE images, with an accuracy of 79% in discriminating benign from malignant lesions and 82% for low-grade versus high-grade tumors. Liem et al. [20] reported optimistic results regarding the association of WLC + CLE, with a high level of sensitivity and specificity for low-grade UC (79% vs. 78%) and similar for HG ones with a slightly higher specificity (67% vs. 79%). From the experience of previous works, the authors outlined the predominant CLE feature to discriminate between LG and HG lesions, namely papillary configuration, organization of cells, cellular morphology, and definition of cell borders. Those characteristics appear easily identifiable and assure a moderate to substantial interobserver agreement, which is mandatory for large-scale CLE application. However, the reported concordance of CLE evaluation by three observers and histopathology was only 63.6%.

### 4.2. CLE Advantages for BCa Management

One of the principal limitations of WLC is the inability to directly discriminate hyperemic flat areas suspicious for carcinoma in situ (CIS), which can be missed or confounded with inflammation. CLE provides clear advantages for those situations, as shown in the examined papers. The reliability of CLE imaging has been provided in the studies reported in this SR. Wu et al. [18] reported that the correspondence of CLE imaging plus WLC evaluation corresponded to the final histological diagnosis in 81% of cases, even without a previous clinical experience on CLE utilization.

Another important application of CLE relates to the detection of carcinoma in situ (CIS) in lesions, which are often misdiagnosed by WLC. The only work that addresses this argument is the one by Lee et al. [17] in which the authors reported the comparison between 21 CIS cases analyzed by CLE and the subsequent biopsies, compared to a similar group of "non-CLE" ones, reporting a CIS detection rate of 83.3%, which is almost 20% higher than on WLC alone; it is also important to state that CIS microscopic features are variable, as a cellular layer can be completely flat or present a thickening of the urothelium. This can bias image interpretation; therefore, a specific formation on CLE image interpretation is also crucial to optimize the CLE detection rate. According to the authors of [17], the CLE aspects of CIS appeared presented more pleomorphic and large cells compared with inflammatory lesions with extensive acellular areas and indistinct cell borders. However, CIS cases are not reported on further in other series, and this should be a main theme for future investigations.

Moreover, CLE can provide significant advantages in the diagnostic assessment of BCa, allowing a clear and real-time recognition of small and low-grade tumors that can be treated by office fulguration instead of needing a surgical retrieval and histologic exam. This statement is outlined in the paper by Liem et al. [20]. According to those authors, this could potentially shift the treatment of those lesions, with a reduction in surgical load, waiting lists for surgery, and medical costs for hospitalization, also avoiding the potential complications of TURBT. This is also outlined as the greatest advantage of CLE in the paper by Beji et al. [21], even if the authors outline that further improvements in imaging processing are still needed to address this statement (see CLE Limitation Section 4.3). In fact, CLE-pooled sensitivity and specificity for low-grade urothelial carcinomas resulted 0.72 and 0.87, respectively, while it is 0.82 and 0.84, respectively, for high-grade ones. Thus, the false-negative rate for high-grade lesions is still too high to make this technique comparable to a histologic report, as there is the risk of missing potential high-grade tumors.

However, CLE can have a role in verifying surgical radicality after TURBT, ensuring the presence of detrusor muscle in the resection specimens with clear benefit for the patient and reducing the needs for an early repeated TURBT.

Another possible application of CLE is in combination with other optical imaging techniques like PDD, NBI, or OCT in view of a multimodal optical assessment. Even if interesting, the feasibility of combining two or more techniques is limited by the technical requirements (need of different cystoscopes, light filters, probes, and devices) [31]. However, there are already different reports in the literature that introduce the concept of "multiparametric cystoscopy" combining the information delivered from the different diagnostic tools available and validating it for clinical use. Kriegmar et al. [32] tested real-time multispectral imaging combining the imaging derived from WLC, PDD, and different enhanced vascular imaging technologies in 31 patients with suspected BCa. Another important study in this field is the one by Marien et al. [33] where the authors evaluated the feasibility of using CLE with two fluorophores, fluorescein and hexylaminolevulinate (HAL), and comparing two Cellvizio probes. The real-time evaluation of tissues reported a sensitivity and specificity of this association of 80 and 100%, respectively, compared to a histologic report. In fact, CLE evaluation was restricted to HAL-suspected areas, therefore maximizing the tumor detection rate. Moreover, the authors outline that the specificity was strongly related to the contrast used; thus, new forms of fluoresceine will improve the diagnostic accuracy of the methodic. This opens the way to a new era for the diagnostic imaging of BCa in line with the most recent technological developments.

### 4.3. CLE Limitations

Even if promising, CLE is still a new-coming device in clinical practice and only three papers [18–20] report a series including up to 50 patients. Moreover, only two papers have been published in the last three years. In fact, five papers were already present before the publication by Wu et al. of the SR [34] with the same argument. This can be partially explained by the advent of the COVID-19 pandemic, which has slowed down many clinical

activities, but it is also related to the actual limitations of CLE imaging for BCa diagnosis. The first is a technical one intrinsic to the probe, as the CLE probe needs a perpendicular contact with the bladder mucosa for image caption and delivering, which is not possible for lesions located in the anterior zone of the bladder [16]. Moreover, the probe is subjected to motion artefacts related both to the patient or the operator, and this can impair the quality of registration.

Afterward, CLE imaging needs a correct interpretation, and this could create a bias among readers, as stated before. This suggests the need of structured training before starting CLE image interpretation in a clinical practice. Moreover, a greater amount of data regarding the CLE imaging of bladder tumors should be provided by multicenter studies to further validate image assessment and interpretation.

The recent study by Beji et al. [21] outlines that even if CLE in its actual form has a high negative predictive value (81% in this study), it is not still capable of being a potential substitute of histologic examination, especially for HG tumors, where a misdiagnosis can negatively affect the natural history of the disease. Therefore, further studies are needed to implement CLE accuracy and imaging interpretation before it can be presented as a possible alternative to histologic examination of retrieved specimens after TURBT. Specific tutoring through video sessions and real-time assessment tests are therefore further encouraged to improve the CLE image interpretations, as stated in the analyzed studies. Another potential solution, according to Lucas et al. [17], can be the validation of a dedicated software able to automatically acquire CLE imaging, delivering a video sequence of the bladder mucosa, which can allow an easier and more correct interpretation of CLE in contrast to a single image. This can reduce interobserver interpretation variability; however, it can also be potentially biased by software limitations.

Other limitations of the application in clinical practice are the cost-effectiveness of the method in comparison to WLC and the availability of CLE probes, which limit CLE's current application only to clinical trials with defined protocols. However, a potential reduction in misdiagnosis through improved diagnostic accuracy could easily overcome this statement and this has already been documented for other diseases in the literature [35]. Thus, we predict a constant increase in CLE imaging adoption in clinical settings in the near future, in parallel to the technical amelioration of fluorescent contrast and probe technology.

## 5. Conclusions

From the reported analysis of the existing literature, CLE appears to be a promising tool that can improve BCa detection and management as a step forward in relation to conventional cystoscopy. In fact, this system can allow the detection of flat lesions and assess the completeness of resection and, in the meantime, provide evidence of low-grade lesions that can be potentially managed in ambulatory settings. This technology has the potential to shift the urologic approach to BCa from only a macroscopic ("visible") to a microscopic ("nonvisible") diagnostic pathway, bringing forward the diagnosis of flat lesions and reducing the rate of false-negative cystoscopies or incomplete TURBT. However, there are still technical limitations, and the actual detection rate still does not appear to allow the safe evaluation of a lesion without comparison to the final histological report. Due to the actual paucity of data, further research is encouraged to overcome the technical limitations of the machine and to optimize image interpretation. This will lead to increased interest among clinicians over this new technology, with relevant benefit for clinical outcomes.

**Author Contributions:** Conceptualization, G.M.P. and A.N.; methodology, G.M.P.; validation, A.G. and A.N.; formal analysis, A.N.; investigation, G.M.P.; resources, A.G.; data curation, G.M.P.; writing—original draft preparation, G.M.P.; writing—review and editing, A.N.; visualization, A.N.; supervision, A.G. All authors have read and agreed to the published version of the manuscript.

**Funding:** This research received no external funding.

**Institutional Review Board Statement:** Not applicable.

**Informed Consent Statement:** Not applicable.

**Data Availability Statement:** Not applicable.

**Conflicts of Interest:** The authors declare no conflict of interest.

## Abbreviations

| | |
|---|---|
| BCa | bladder cancer |
| CIS | carcinoma in situ |
| CLE | confocal laser endomicroscopy |
| HAL | hexylaminolevulinate |
| HG | high-grade |
| LG | low-grade |
| NBI | narrow-banding imaging |
| NMIBC | non-muscle invasive bladder cancer |
| OCT | optical coherence tomography |
| PDD | photodynamic diagnosis |
| PRISMA | preferring reporting items for systematic reviews and metanalysis |
| QUADAS | quality assessment of diagnostic accuracy studies |
| SPIES | Storz professional image enhancement system |
| UTUC | upper tract urothelial carcinoma |
| WLC | white-light cystoscopy |

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
