# Peer review of "Confocal Laser Endomicroscopy for Bladder Cancer Detection: Where Do We Stand?"

_applsci, doi:10.3390/app12199990_

Round 1
Reviewer 1 Report
Comments to the Author
This paper entitled “Confocal Laser Endomicroscopy for bladder cancer detection. Where do we stand?” by Angelo Naselli, Andrea Guarneri, Giacomo Maria Pirola is interesting as CLE appears a promising tool which can improve BCa detection and management as a step forward in relation to conventional cystoscopy. This technology has the potential to shift the urologic approach to BCa form an only macroscopic (“visible”) to a microscopic (“nonvisible”) diagnostic pathway, bringing forward the diagnosis of flat lesions and reducing the rate of false-negative cystoscopies or incomplete TURBT.
The authors need to summarize their evaluation of this technology in an objective, quantifiable, and easy-to-understand manner.
Comments
1: Table 1 only lists Low High, but should be quantified or graded ratings should be included for each paper whenever possible.
Criteria for criteria should be clearly defined and reviewed.
2. Table 2 is too chaotic. Reports without sufficient information do not seem appropriate for review.
3. The authors did not fully examine pathological and other evaluations in this review. In cancer, pathological evaluation, such as the original histological type and degree of differentiation, is very important for the outcome of various technological treatments, but the authors do not provide figures such as what tissues were used for the evaluation. Even if this is a review, the evaluation should be mentioned.
Reviewer 2 Report
Angelo Naselli et al. summarized seven prospective studies of CLE in bladder cancer systematically and concluded it as a promising tool for diagnosis. They pointed out the advantages and disadvantages in detail. However, the analysis looks a little bit simple, including the result description, despite the limitations on the number of literature.
Please address the below concerns:
1. Line 15, “both from a diagnostic than from a clinical point of view” it’s “and” instead of “than”?
2. Line 17, “systematic review (SR)” move this full name to line 16.
3. Line80, delete space in pub med.
4. Line 24 and Line 112, the numbers of patients and bladder tissues, not consistent
5. Please see the similar work of search in Jan 2020 paper “Diagnostic Performance of Confocal Laser Endomicroscopy for the Detection of Bladder Cancer: Systematic Review and Meta-Analysis”. In this manuscript review, is updated to August 2022. However, it seems very few new papers followed this field in recent two years, is this really a promising direction? Please explain. Maybe try more search with words like urothelial carcinoma, or urinary. And this should be in the citation or discussion, because it’s so close to the current manuscript.
6. Suggest more descriptions of Table 2. Please insert Table, instead of a figure in Table 2. More analyses in the results part are recommended.
7. Line 245 font format is not correct. Manuscript including introduction, method, results and discussion parts has some problems with line spacing.
Round 2
Reviewer 1 Report
The manuscript was improved.
Author Response
Thanks a lot for the kind appreciation and for the time spent for revision, Our best regards.
Reviewer 2 Report
The authors addressed several concerns. However, still have four left, please see below.
1. Line 24 and line 134: why are patient numbers overall still not the same (214 or 241)? It's confusing, or have some patients been excluded? Even that, you still have to make a clear statement in the results part with exact same patient numbers shown in the abstract. Please double checking the calculations to give the right numbers of patients and lesions.
2. About remaking a Table 1, didn't get the point for the response, just formatting this easy table to fit the page, figure and table should be same size in the text. I would highly recommend using the original table, otherwise figure in the manuscript is realy low-resolution, even worse than the one in the first submission, and figure even has typing errors signs in it (red wavy underline), Can't imagine.. So one more error for the Correspondence spelling.
3. I did see the authors try to make a better presentation on Figure 2 using previous table. However, it just looks worse in my opinion, suggest to just use previous one, because only 5 samples in group, don't have to make a figure of propportion, make it simple. If author try to keep this figure, it's fine, but please make it look better and comfortable, no red wavy undeline in figure.
4. In discussion part, the paragraph spacing is just not right. Other parts still have some spacing problems or need to justify text to both sides. I will leave this to the journal publishing or proofreading to figure this out, if authors couldn't fix it.
